# Gold Nanoparticles Synthesized with Common Mullein (*Verbascum thapsus*) and Castor Bean (*Ricinus communis*) Ethanolic Extracts Displayed Antiproliferative Effects and Induced Caspase 3 Activity in Human HT29 and SW480 Cancer Cells

**DOI:** 10.3390/pharmaceutics14102069

**Published:** 2022-09-28

**Authors:** Karen M. Soto, Ivan Luzardo-Ocampo, José M. López-Romero, Sandra Mendoza, Guadalupe Loarca-Piña, Eric M. Rivera-Muñoz, Alejandro Manzano-Ramírez

**Affiliations:** 1Centro de Investigaciones y de Estudios Avanzados del I. P. N. Unidad Querétaro, Queretaro 76230, Mexico; 2Instituto de Neurobiología, Universidad Nacional Autónoma de México (UNAM-Campus Juriquilla), Queretaro 76230, Mexico; 3Research and Graduate Program in Food Science, Universidad Autónoma de Querétaro, Queretaro 76010, Mexico; 4Centro de Física Aplicada y Tecnología Avanzada (CFATA), Universidad Nacional Autónoma de México (UNAM-Campus Juriquilla), Queretaro 76230, Mexico

**Keywords:** antiproliferative effect, castor bean (*R. communis*), colorectal cancer, common mullein (*V. thapsus*), gold nanoparticles, reactive oxygen species

## Abstract

Gold nanoparticles (AuNPs) are promising nanomaterials exhibiting anti-cancer effects. Green AuNPs synthesis using plant extracts can be used to achieve stable and beneficial nanoparticles due to their content of bioactive compounds. This research aimed to synthesize and evaluate the antiproliferative and caspase-3 activity induction of green AuNPs synthesized with common mullein (*V. thapsus*) flowers (AuNPsME) and castor bean (*R. communis*) leaves (AuNPsCE) ethanolic extracts in human HT29 and SW480 colorectal cancer cells. Their effect was compared with chemically synthesized AuNPs (AuNPsCS). The extracts mainly contained *p*-coumaric acid (71.88–79.93 µg/g), ferulic acid (19.07–310.71 µg/g), and rutin (8.14–13.31 µg/g). The obtained nanoparticles presented typical FT-IR bands confirming the inclusion of polyphenols from *V. thapsus* and *R. communis* and spherical/quasi-spherical morphologies with diameters in the 20.06–37.14 nm range. The nanoparticles (20–200 µg/mL) showed antiproliferative effects in both cell lines, with AuNPsCE being the most potent (IC_50_ HT29: 110.10 and IC_50_SW480: 64.57 µg/mL). The AuNPsCS showed the lowest intracellular reactive oxygen species (ROS) generation in SW480 cells. All treatments induced caspase 3/7 activity to a similar or greater extent than 30 mM H_2_O_2_-treated cells. Results indicated the suitability of *V. thapsus* and *R. communis* extracts to synthesize AuNPs, displaying a stronger antiproliferative effect than AuNPsCS.

## 1. Introduction

Gold nanoparticles (AuNPs), also called “gold colloids”, are usually particles with different morphologies, including nanospheres, nano bars, nanolayers, and nanocubes, and small sizes ranging from 1 to 100 nm. These nanoparticles represent one of the most promising nanomaterials because of various advantages, such as their ease of synthesis, bio-molecular coupling abilities or easy functionalization with almost every type of electron-donating molecule, and multiple properties, allowing their use in different applications, e.g., sensors, photodynamic therapies, bioimaging, drug carriers, and others [1,2,3,4]. 

One of the most important applications of AuNPs is their use as therapeutic molecules for the medical treatment of cancer. Gold nanoparticles exhibit cytotoxic properties against certain types of cancer cell lines like Breast cancer cells (MDA-MB-231) [5], liver cancer cells (HepG2) [6], pancreatic cancer cells (PANC-1) [7], and others. The mechanism of action of the cytotoxic properties has not yet been fully described so far. However, some of the possible mechanisms include interaction with the cell membrane and internalization in the cell by different charges on the surface of cells and AuNPs showing cytotoxic activity via ROS (Reactive Oxygen Species) production, DNA and mitochondrial damage, activation of caspase cascade of apoptosis, cell rupture and inhibition of cell growth, and loss of membrane stability [8,9,10]. Colorectal cancer (CRC) is a common malignancy and one of the leading causes of cancer death worldwide; the second and third most frequently occurring cancer in women and men, respectively, is the third most common cancer, with 1.8 million new cases diagnosed in 2018. Colon cancer cell lines, e.g., HT29 and SW480, are fundamental tools for analyzing the role of antitumor compounds and treatments [11,12].

Several methods can be used to synthesize AuNPs. The most used is the synthesis involving the reducing of a precursor agent (HAuCl_4_) with reducing agents, such as sodium borohydride (NaBH_4_). However, generating toxic compounds harmful to humans and the environment requires searching for new sources of reducing agents. In this sense, plant extracts represent a suitable alternative with multiple advantages over other green synthesis methods. Plant extracts have a high quantity of diverse phytochemicals, such as proteins, polyphenols, and related multifunctional-electron-rich species serving as dual reducing and stabilizing agents, encapsulating gold atoms through the phytochemical crown on the AuNPs surface, increasing their stability [10,13]. Moreover, AuNPs synthesis does not need microbial cultures, making them more straightforward to further scale-up.

Considering the wide variety of medicinal plants, our research group has previously examined underutilized plant species and derived wastes for manufacturing nanoparticles [10,14]. Underutilized plant species, castor bean (*R. communis*), and common mullein (*V. thapsus*) are traditionally used to treat several conditions. Castor bean is a plant found in various regions of the world, mainly in tropical areas, and is widely used to produce castor oil. Castor bean leaves have been used as a medicinal plant due to the presence of phytochemicals, mainly phenolic compounds such as flavonoids (rutin, quercetin, and (+)-catechin), hydroxycinnamic acids (caffeic, chlorogenic, *p*-Coumaric, and sinapic acids), and hydroxybenzoic acids (gallic and ellagic acids) [15]. As a result of these compounds’ presence, the leaves and leaves extracts have been used for their anti-inflammatory [16] and anti-cancer effects [17]. On the other hand, common mullein (*V. thapsus*) is a plant used in traditional medicine to treat inflammatory diseases, bronchial congestion, coughs, diarrhea, and snakebite [18]. *R. communis* leaf extract was used for the green synthesis of spherical gold nanoparticles with diameters between 40 and 80 nm and probed against Hela and HepG2 cells, obtaining 58.64% and 42.74% inhibitory effects, respectively [19], while *V. thapsus* was used for the synthesis of metallic nanoparticles, e.g., AgNPs for antibacterial activity [20], CuO nanoparticles for photocatalytic degradation of methylene blue [21], and Iron nanoparticles for Cr (VI) reduction [22].

As one of the most critical applications of green gold, nanoparticles are used for their anti-cancer effects, and plant extracts contain bioactive compounds displaying anti-cancer benefits and are functionally optimal as reducing agents to synthesize stable AuNPs. Their green synthesis using either castor bean or common mullein provides a technological activity with medicinal value against several types and stages of cancer [23,24]. Therefore, this research aimed to synthesize, characterize, and evaluate the antiproliferative effect of green AuNPs made with castor bean leaves and common mullein flowers’ ethanolic extracts against two cell lines of colorectal cancer, one of the most critical sicknesses. It is important to mention that this is the first report of the use of *V. thapsus* for green gold synthesis and the first report of the antiproliferative against CRC effect of *R. communis* gold nanoparticles. In this study, the AuNPs obtained with chemical synthesis and green synthesis were compared to observe the differences that may occur in the biological activity and in the mechanism of antiproliferative action, which could be affected by the extracts present in the plants. It is important to mention that this is the first report of the use of *V. thapsus* for green gold synthesis and the first report of the antiproliferative effect of *R. communis* gold nanoparticles against CRC.

## 2. Materials and Methods

### 2.1. Chemicals and Reagents

Solvents, chloroauric acid (HAuCl_4_), sodium; 1,1-diphenyl-2-picrylhydrazyl (DPPH); 2,2′-azinobis-(3-ethylbenzothiazoline-6-sulfonic acid) (ABTS), NaBH_4_, 2′,7′-dichlorodihydrofluorescein (H_2_DCFDA), H_2_O_2_, DMSO, HPLC standards, and potassium persulfate were purchased from Sigma Chemical Co. (St. Louis, MO, USA). Dulbecco’s Modified Eagle’s Medium (DMEM), fetal bovine serum (FBS), and antibiotic antimycotic solution were from Gibco, Waltham, MA, USA. The EnzChek^®^ Caspase-3 Assay was purchased from ThermoFisher Scientific (Waltham, MA, USA).

### 2.2. Extract Obtention and Characterization

Common mullein (*V. thapsus*) (World Checklist of Selected Plant Families, WCSP record in review: Sp. Pl. 177 1753) flowers and castor bean (*R. communis*) (WCSP record: 178867) leaves were collected in Queretaro (Mexico) and deposited in the “Dr. Jerzy Rzedowski” (QMEX) herbarium of Universidad Autónoma de Querétaro (Juriquilla, Querétaro, Mexico). The selected plant parts were washed with deionized water and dried in a convection oven at 60 °C for 6 h. The dried residues were then powdered to fine particles using a mixer grinder (KRUPS GX4100, Solingen, Germany) and passed through a 60-mesh screen (250 µm particle size). Ten grams of the powdered castor and mullein plants were added to 100 mL ethanol and left to stand for 4 h. The extracts were filtered (Whatman Paper no. 40) and stored at 4 °C for further analysis. These extracts were termed ME and CE for common mullein and castor bean, respectively.

To spectrophotometrically analyze phenolic compounds, total phenolic compounds (TPC) were determined using the Folin-Ciocalteu technique [25], and total phenolics were expressed as mg of gallic acid equivalents per gram of dry weight (mg GAE/g DW). For the in vitro antioxidant capacity (AOX), the DPPH [26] and ABTS^•+^ [27] radical scavengers were tested, and the results were indicated as mg Trolox equivalents per gram of dried weight (mg TE/g DW).

The identification and quantification of individual phenolic compounds were performed through high-performance liquid chromatography coupled to a diode array detector (HPLC-DAD) [28]. Briefly, an Agilent 1100 Series HPLC System (Agilent Technologies, Palo Alto, CA, USA) with a DAD detector (Agilent Technologies), and using a thermostatically controlled (35 ± 0.6 °C) Zorbax Eclipse XDB-C18 column (Agilent Technologies, 9.4 × 250 mm, 5.0 μm of granule size) was used. The samples were eluted using a linear gradient of concentration from water containing 0.1% of acetic acid (Solvent A) to 100% of acetonitrile (solvent B) as follows: 80–83% A for 7 min, 83–60% A for 5 min, 60–50% A for 1 min, and 50–85% A for 2 min (The solvent B complete the 100%). An acquisition speed of 1 min with a volume injection of 20 µL was used, and the samples were analyzed in triplicate. The quantification was carried out using external HPLC-grade chemical standards, and curves were plotted. Compounds were expressed in mg equivalents of each one per gram of dry weight (mg eq/g DW).

### 2.3. Green (AuNPsME and AuNPsCE) and Chemical (AuNPsCS) Synthesis of Gold Nanoparticles

For the chemical synthesis, AuNPsCS were prepared with the methodology presented by Deraedt et al. [29]. For this, 1.5 mg of HAuCl_4_ (Mw = 339.7 g/mol, n = 4.4 × 10^−3^ mol) were dissolved in 32 mL of water to obtain [Au] = 1.38 × 10^−1^ mM. After 15 min of stirring, 1 mL of a water solution of NaBH_4_ (10 equivalents of Au) was added quickly.

For the green synthesis of AuNPs, 10 mL of an aqueous solution of HAuCl_4_ (0.5 mM) were mixed with 50 and 100 μL of the ethanolic extract of mullein flowers and castor leaves, referred to as AuNPsME and AuNPsCE, respectively. The reaction mixture was incubated at 60 °C for 5 min, the reduction of Au^3+^ ions was monitored using color changes, and the UV−vis spectrum was recorded from 300 to 800 nm with baseline correction using water as the blank on a Spectra Max Tunable Microplate Reader (Molecular Devices Co., Sunnyvale, CA, USA).

### 2.4. AuNPs Characterization

The morphology and diameters of AuNPs were determined using a scanning transmission electron microscope (STEM SU8230, Hitachi, Tokyo, Japan). The diameters of the AuNPs were obtained by measuring at least 50 particles with Image J software [30]. DLS and Z potential were performed using a Zetasizer Nano ZS (Malvern Panalytical, Worcestershire, UK). Infrared absorption spectra of dried AuNPs and lyophilized extracts were recorded using Fourier-transformed infrared (FT-IR) equipment (Spectrometer Spectrum Two, PerkinElmer, Waltham, MA, USA). The X-ray pattern of the AuNPs was obtained with an X-ray diffractometer (Dmax 2100 Rigaku Americas, The Woodlands, TX, USA) that has a CuKα radiation generator (k = 1.5418° A), from 5 to 50 °, on a 2θ scale, with a step size of 0.02° [31]. The thermal analysis of the AuNPs was carried out using a TGA-50, Shimzadu thermogravimetric analyzer (Shimzadu Corp., Kyoto, Japan) by increasing the temperature from 25 to 800 °C at the rate of 5 °C/min, and nitrogen gas flow rate of 10 mL/min.

### 2.5. Cell Culture

Human HT-29 (ATCC HTB-38) and SW-480 [SW-480] (ATCC CCL-228) colon adenocarcinoma cells were acquired from American Type Culture Collection. Cells were cultured using Dulbecco’s Modified Eagle’s Medium (DMEM, Gibco, Waltham, MA, USA) supplemented with 2.2 g/L NaHCO_3_, 10% fetal bovine serum (FBS, Gibco), and 1% antibiotic antimycotic solution (100×, Gibco). The cells were maintained in a humidified 5% CO_2_ atmosphere at 37 °C.

#### 2.5.1. Cell Viability Assay by 3-(4,5-Dimethylthiazol-2-yl)-2,5-Diphenyl Tetrazolium Bromide (MTT) Assay

The assessment of cell viability was conducted using the MTT assay. The cells (3.125 × 10^4^ cells/cm^2^) were seeded in 96-well plates for 24 h. The media (DMEM + 10% FBS) was then replaced with several concentrations of the green gold nanoparticles AuNPsCE and AuNPsME (20, 50, 100, 150, and 200 µg/mL AuNPs) dissolved in DMEM + 0.5% bovine seric albumin (BSA). To avoid excessive cell growth and maintain a similar cell population that was present at the beginning of the experiment. After 24 h, a 5 mg/mL MTT solution dissolved in PBS 1× was added, and the cells were left to stand for 3 h. The media was then replaced with a sodium dodecyl sulfate (SDS)-HCl solution (14 g SDS dissolved in 140 mL distilled water + 112 µL HCl), and the plate was placed in a rocket shaker overnight (40 cycles/min). The absorbance was read at 595 nm in a Varioskan plate reader (ThermoFisher Scientific, Waltham, MA, USA), and the cell viability was expressed as a percentage of the negative control cells (cells without treatments: DMEM + 10% FBS). Cells treated with 30 µM H_2_O_2_ were used as the positive control. The half inhibitory concentration (IC_50_) was calculated using a biological model adjusted equation provided by GraphPad Prism v. 8.2 (Dotmatics, Boston, MA, USA). Three independent experiments using six replicates were used.

#### 2.5.2. Intracellular Reactive Oxygen Species (ROS) Quantification

The cells (2.50 × 10^4^ cells/cm^2^) were seeded in 96-well plates for 24 h. Afterward, the media was replaced with several concentrations of the treatments (20, 50, 100, 150, and 200 µg/mL AuNPs) prepared as indicated in Section 2.5.1., for 24 h [32,33]. Once the media was removed, the cells were then washed twice with PSB 1×, and 100 µM 2′,7′-dichlorodihydrofluorescein (H_2_DCFDA) solution (D399, ThermoFisher Scientific, Waltham, MA, USA), prepared as indicated by the manufacturer in DMSO, was added (final DMSO concentration: 5%). Subsequently, the cells were incubated at 30 min at room temperature (25 ± 1 °C). The fluorescence was read at 485/525 nm (excitation/emission) in a Varioskan plate reader (Thermo Fisher Scientific, Waltham, MA, USA) [34]. The results were expressed as relative fluorescence units (RFU). DMEM + 10% FBS and 30 µM H_2_O_2_-treated cells were used as positive and negative controls, respectively. Two independent experiments in triplicates were conducted.

#### 2.5.3. Caspase 3/7 Activity Assay

The EnzChek^®^ Caspase-3 Assay kit #2 (E13183, Invitrogen™, ThermoFisher Scientific, Waltham, MA, USA)) was used to assess apoptosis detection by assaying increases in the caspase-3 and other Asp-Glu-Val-Asp (DEVD)-specific protease activities such as Caspase 7. The cells (1.05 × 10^5^ cells/cm^2^: 1 × 10^6^ cells/well in 6-well plates) were seeded for 24 h. Then, the media was replaced with 1 mL of the IC_50_ concentration of each extract (IC_50_ AuNPsME: 116.7 and 153 µg/mL; IC_50_ AuNPsCE: 110.10 and 64.57 µg/mL, and IC_50_ AuNPsCS: 151.80 and 126.30 µg/mL, for HT29 and SW480 cells, respectively) for 24 h. Subsequently, the media was removed, cells were washed twice with PBS 1×, trypsinized (0.25% Trypsin-EDTA, ThermoFisher Scientific) for 15 min, and trypsin activity was stopped with 2% FBS-PBS 1×. Then, the cells were centrifuged to collect them (350× *g*, 10 min), and frozen at −80 °C. To quantify the Caspase-3 activity, the manufacturer’s instructions were followed using the collected cells. A caspase-3 inhibitor provided by the kit was also used to treat the cells (1 mM Ac-DEVD-CHO) to confirm that the observed results were due to caspase-3 activity and no other proteases (results not shown). The fluorescence was measured at 496/520 nm of excitation/emission fluorescence, and results were expressed in RFU. Untreated cells and 30 mM H_2_O_2_-treated cells were used as negative and positive controls, respectively. Two independent experiments in duplicates were conducted for each treatment.

### 2.6. Statistical Analysis

Unless indicated, the results were expressed as the mean ± SD of at least two independent experiments in duplicates. An analysis of variance (ANOVA) followed by a post-hoc Tukey–Kramer’s test was used, considering significance at *p* < 0.05. The analysis was carried out using JMP v. 16.0 software (SAS, Cary, NC, USA).

## 3. Results and Discussion

### 3.1. Extraction and Characterization of Ethanolic Extracts

The total phenolic compounds, the antioxidant capacity (ABTS and DPPH radical scavenging), and the individual phenolic composition attained using HPLC-DAD are shown in Figure 1. The best antioxidant activity was obtained with castor bean ethanolic leave extract (CE) with 726.65 ± 10.72 and 429.89 ± 2.01 mg TE/g DW for the ABTS^•+^ and DPPH radical scavenging, respectively, being almost ten times higher compared to mullein extract (Figure 1A). These results could be closely related to the total content of total phenolic compounds (TPC), which are known to provide antioxidant activity. Hence, the castor extract presented the highest range of total phenolic compounds with 101.32 ± 0.81 mg GAE/g DW. The obtained ME-TPC was consistent with previous reports of *V. thapsus* aerial and flower extracts (44.84–60.13 mg GAE/g DW or mL extract) [35,36], although Taleb et al. [36] did not specify the amount of powdered extract used, and the flowers were subjected to several ethanolic extractions (96% *v*/*v*) by maceration.

Castor bean ethanolic extracts also yielded a higher TPC (Figure 1A) than reported values for methanolic and aqueous extracts of castor bean leaves (~30–50 mg GAE/g DW) [15]. The values are significantly lower than TPC contents from five *R. communis* Tunisian populations (174.25–623.7 mg GAE/g DW) [37]. However, the authors conducted a methanol-maceration extraction that was further concentrated in a rotary evaporator, which could explain almost 10-times the TPC than this study.

It has been reported that the DPPH and FRAP antioxidant capacities from *V. thapsus* aqueous and ethyl-acetate extracts are positively correlated with (+)-quercetin, rutin, ferulic acid, harpagoside, protocatechuic acid, rosmarinic acid, and salicylic acid, similarly to some of the identified and quantified compounds found in ME [35]. The richness in phenolic compounds of *V. thapsus* exhibiting a high ABTS^•+^ and DPPH radical scavenging was demonstrated by Mihailović et al. [38], which showed that *V. thapsus* methanolic (8.11% *v*/*v* methanol) and aqueous (10.03% water) extracts from their aerial parts displayed some of the lowest IC_50_ values of radical scavenging in comparison with other *Verbascum* species. Considering CE, polar solvents have been validated as some of the most efficient extracting antioxidant compounds from *R. communis* leaves, displaying a high antiradical activity by ABTS^•+^ and DPPH methods [39].

The identification and content of 13 phenolic compounds conducted using HPLC-DAD are presented in Figure 1B as a composition heatmap. Three hydroxybenzoic acid derivatives and benzaldehydes, four hydroxycinnamic acids and derivatives, two flavonoids, three flavones, and one hydrolyzable tannin (Appendix A) were identified and quantified in the extracts. The extracts presented different concentrations and very noticeable differences in certain compounds. The main compounds in ME were *p*-coumaric acid, gallic acid, (+)-catechin, ferulic acid, and rutin. Mihailović et al. [38] reported similar caffeic acid concentrations from methanolic and water *V. thapsus* extracts (0.81–1.53 µg/g DW). Selseleh et al. [35] reported lower concentrations of *p*-coumaric acid (0.11 µg/g DW), ferulic acid (1.33 µg/g), rutin (2.90 µg/g), and similar levels of quercetin (1.35 µg/g) from methanolic fractions of *V. thapsus* flowers.

Regarding CE, the main compounds are ferulic acid, epigallocatechin gallate, coumaric acid, and caffeic acid (Figure 1B, Appendix A). Vasco-Leal et al. [15] reported similar contents of gallic acid (10–25–35.81 µg/g DW), *p*-coumaric acid (0.63–517.12 mg/g DW), rutin (1.04–5.91 µg/g DW), and (+)-catechin (6–147.65 µg/g) than those found in this study. Caffeic acid was positively correlated (*p <* 0.05) with DPPH radical scavenging in *R. communis* extracts. In both samples, phenolic acids, whose structure is shorter than flavonoids, are presented as major compounds, and their possible participation in the synthesis and stabilization of metallic nanoparticles such as silver has been verified [40].

### 3.2. Synthesis and Characterization of AuNPs

The synthesis of AuNPs was monitored with color change and UV-Vis spectra. Figure 2A shows the UV-Vis spectra of AuNPs synthesized with the ethanolic extracts and NaBH_4_ (chemical synthesis: AuNPsCS). AuNPsME presented a red wine color and a maximum absorption band of 550 nm. AuNPsCE showed a red wine coloration after the synthesis and two absorption bands at 540 nm and 675 nm. The synthesis of particles can explain these two bands with different diameters or nanorods. In contrast, the AuNPsCS showed a purple hue and maximum absorption at 548 nm. Generally, metal nanoparticles have free electrons, which give surface plasmon resonance (SPR) absorption band due to the combined vibration of electrons of metal nanoparticles in resonance with a light wave. The sharp bands of gold colloids for the plasmon resonance appeared in the visible range at 540–550 nm, dependent on the morphology and size distribution [41]. Rahman et al. [42] obtained dark brown AuNPsCE made with 5-fold more concentrated CE than those reported in this research. Moreover, an absorption peak at 550 nm was observed, close to the reported 540 nm from this research.

#### 3.2.1. Thermogravimetric Analysis

Thermogravimetric analysis (TGA) is presented in Figure 2B. TGA was conducted to determine the phytochemical compounds (phenolic and small proteins) that capped the surface of AuNPs. Green gold nanoparticles synthesized with mullein and castor extracts showed a two-stage process of losing weight. AuNPsME had a 23% weight loss in a 221–455 °C range and a total weight loss of 36% at 800 °C. AuNPsCE presented a weight loss of 27% in the 115–460 °C range and a final weight loss of 53.2%. The first weight loss can be attributed to the capped organic materials on the surface of particles. After 400 °C, the degradation of resistant aromatic compounds present in the extract compounds is presented, and the final weight loss refers to the presence of pure AuNPs [43,44]. In chemical synthesis, only one stage was observed at 305–800 °C, with a final weight loss of 83.3%. These results indicate that the bioactive compounds present in the extract, principally polyphenolic compounds, were capped on the synthesized AuNPs and were degraded due to the high temperature, which is not observed in chemically synthesized AuNPs.

#### 3.2.2. Infrared Spectroscopy

Figure 2C presents the FT-IR spectra of pure ethanolic mullein, castor extracts, and synthesized gold nanoparticles. The FT-IR method was used to identify the responsible functional groups involved in reducing, synthesizing, and capping the gold nanoparticles. All the spectra exhibited similar bands: between 3800–3200 cm^−1^ can be assigned to the phenolic hydroxyl and aliphatic hydroxyl groups (-OH); bands in 2900–2600 cm^−1^ correspond to symmetric stretching of methyl (CH_3_) and methylene (CH_2_) group; absorption bands in 1760–1730 cm^−1^ indicate C=O stretching, suggesting the presence of high quantities of phenolic acids; the band at 1600–1580 cm^−1^ represents the presence of C=C stretching in aromatic compounds; bands in 1390–1350 cm^−1^ indicate alkane (C-H) bending and asymmetric stretching of the glycosidic bond (C-O-C), and the 700–600 band corresponded to the CH out of plane bending in carbonyl compounds [15,42,45]. Overall, the fingerprint region between 800 and 1200 cm^−1^ indicates the area for carbohydrate-based biopolymers. The band corresponding to the hydroxyl groups became narrower after synthesis, suggesting that hydroxyl groups play an important role in gold ion reduction and the formation of hydroxyquinones that capped the AuNPs surface [46,47].

#### 3.2.3. Morphology and Size Distribution

The morphology and size distribution of the AuNPs was examined by STEM and DLS analysis (Figure 3). Most nanoparticles presented a quasi-spherical morphology, and only in green synthesis exhibited triangular morphologies. AuNPsME mainly presented spherical morphologies with 20.1 ± 0.2 nm. For AuNPsCE, quasi-spherical morphologies with a median diameter of 37.1 ± 0.4 nm predominated. On the contrary, AuNPsCS exhibited spherical and flower-like morphologies with the highest diameters (102.3 ± 2.1 nm). In green synthesis, the particles with triangular morphologies present a size larger than the quasi-spherical morphologies, AuNPsME presents an average diameter of 30.8 ± 12.7, and AuNPsCE of 41.3 ± 16.2. Although the mean diameters are not so different from the other morphologies, they present larger standard errors, indicating a greater dispersion.

Results from green synthesis are consistent because hierarchical structures, such as triangular ones, are formed by smaller particles, generally spherical, considered growth seeds. Due to their size and shape, these structures are less stable for aggregation. On the other hand, flower-like nanoparticles synthesized with NaBH_4_, presented larger diameters corresponding to the aggregation. The growth of nanoparticles may be assisted by the capping agents, limiting the size and controlling the shape. The non-specific binding on all the exposed surfaces of gold in green synthesis may lead to anisotropic growth. In addition, the different compounds present in the plant extracts may prevent aggregation as capping agents [48]. The hydrodynamic diameter of DLS follows a similar trend to STEM diameter as the smallest particles are synthesized with ME with 49.41 nm and a PDI (polydispersity index) of 0.437, followed by the AuNPsCE with 56.2 nm and PDI of 0.391, and the largest are those of chemical synthesis with 92.12 nm and 0.143 of PDI. Generally, the hydrodynamic diameters are more prominent because they include the compounds on the surface and cannot differentiate two particles very close together. The PDI values greater than 0.1 may imply polydisperse particle size distributions, which may be due to the variable compound composition of the extracts that cap the surface of AuNPs [49,50].

Surface Z potential was measured to record the surface charge on green and chemical AuNPs. Generally, nanoparticles are considered to be stable if the surface potential values are between −30 mV and +30 mV [51]. The zeta values of AuNPsME, AuNPsCE, and AuNPsSQ are 25.43 ± 0.66, 34.96 ± 0.79, and 63.86 ± 1.18 respectively. These negative zeta potentials confirmed the formation of negative charges on the surface of the AuNPs and predict that the particles made with green synthesis have greater stability compared to the particles made by chemical synthesis. The anticancer effect of AuNPs was found to be highly dependent on a range of factors related to their physical characteristics, such as surface coating, shape, size, and superficial charge. Regarding the size, particles with small size and negative charge, like that obtained in green synthesis, are capable of taking advantage of the enhanced permeability and retention effect.

### 3.3. Ethanolic Extracts-Based AuNPs Are More Cytotoxic than AuNPsCS

MTT analysis was used to assess the cytotoxic activity of different concentrations of green and chemical AuNPs in two colon cancer cell lines, SW480 and HT29, which vary in terms of epithelial-mesenchymal transition (EMT) markers, WNT activity, and stemness signatures. SW480 is derived from a primary Duke’s stage B adenocarcinoma. It has a high expression of EMT markers, with a migratory capacity phenotype, while the HT29 derived from Duke’s C stage human colon adenocarcinoma cell line has been ranked lower in terms of EMT and humanized intestinal stem cell signature, challenging the system to differentiate between different stages of the same disease [12,52].

#### 3.3.1. Cellular Viability of AuNPsME

Figure 4 shows the impact of AuNPsME on HT29 and SW480 cell viability. AuNPsME extracts >100 µg/mL are needed to decrease HT29 cell viability <80% (Figure 4A), whereas overall lower cell viability was observed in SW480 cells (Figure 4B). However, since a major impact was shown between 100 and 150 µg/mL for HT29, their calculated IC_50_ values were lower than those for SW480 (−23.72%).

Representative pictures for the effect of AuNPsME in HT29 and SW480 cells (Figure 4E,F) showed high levels of cells unsticking at AuNPsME > 150 µg/mL, which coincided with the observed cell viability tests. As indicated by the IC_50_ values and the composition of the extracts shown in Appendix A, the obtained concentrations mainly contained 4.60–8.79 µg/mL p-coumaric acid, 2.10–19.88 µg/mL ferulic acid, 0.57–5.62 µg/mL epigallocatechin gallate, 0.36–2.40 µg/mL caffeic acid, and 1.03–1.73 µg/mL hydroxyphenylacetic acid, among other compounds. Only a few reports have tested the safety of ME in selected models. For instance, Dragičevic et al. [53] suggested that *V. thapsus* flowers aqueous extracts, prepared at 1:10 flower:water maceration, further rotary evaporator-mediated concentration up to 36 µg/mL, and dilutions starting from 1000× (0.036 µg/mL), did not induce malformations in zebrafish embryo, but viabilities in the range of 57.9–67.3% in human lung epithelial cells (BEAS-2B) suggested certain toxic effects that merit further research. Other authors reported that 50–200 µg/mL *V. thapsus* aqueous leaves extracts are safe in mice RAW264.7 macrophages and human adult chondrocytes (HC) [54]. Aerial parts of *V. thapsus* at 100, 300, and 500 mg/kg body weight, which are much higher than the doses used in this research, did not produce any cytotoxic effects expressed in the increase of frequency of micronucleated polychromatic erythrocytes (MNPCE) in BALB/c mice after a 24 h administration [55]. There are no reports on the impact of *V. thapsus* or even *V. thapsus*-based nanoparticles on HT29 and SW480 cells. However, AuNPs made from *V. thapsus* leaf extracts (20 µg/mL) showed IC50 of 287–308 µg/mL against differentiated bronchogenic carcinoma HLC-1 and moderately differentiated lung adenocarcinoma cells [56].

#### 3.3.2. Cellular Viability of AuNPsCE

The impact of AuNPsCE on HT29 and SW480 cell viability is shown in Figure 5. A dose–response trend was observed for both cell lines (Figure 5A,B), with the effects being stronger on SW480 cells since even the lowest concentration showed cell viabilities <80%. The results could be confirmed in the 1.7-fold lower IC50 values for SW480 compared to HT29 cells (Figure 5C,D). The representative pictures showed a high degree of viability loss at the highest concentration for HT29 cells (Figure 5E), whereas results for SW480 cells are similar between 50 and 200 µg/mL (Figure 5F). Previously, our research group tested CE extracts on RAW264.7 macrophages, and concentrations in the range of 10–50% *v*/*v* (equivalents to 2–5 mg/mL aqueous CE) were not cytotoxic (cell viability > 80%) (results not shown) [15]. Gold nanoparticles made from *R. communis* leaf extracts showed inhibition values ranging 30–70% from 25 to 200 µg/mL AuNPsCE on normal rat splenic cells, although no differences were shown from 50 to 200 µg/mL AuNPsCE [42], suggesting that the doses assayed in this research can be considered safe within the 20–100 µg/mL range. Particularly for human hepatic (HepG2) and cervical (HeLa) cell lines, AuNPsCE inhibited 58.64% and 42.74% cell growth, respectively, but no mechanistic effects were provided. Although there are no reports on the effect of AuNPsCE on colon cancer models in vitro or in vivo, CE extracts from several castor bean parts have been assayed in several cancer types. An ethanolic extract of the fruit demonstrated reduced cytotoxicity (0.005–1 µg/mL) in human MCF-7 and MDA-MB-231 breast cancer cell lines [57], and chloroform extracts from the oil demonstrated antiproliferative effects on human SK-MEL-28 skin melanoma cells [58]. Taken together, *R. communis* extracts have increasingly been assayed for their anticancer effects, and encapsulation methods aiming to protect its bioactive molecules ensure an enhanced effect against cancer cells.

#### 3.3.3. Cellular Viability of AuNPsCS

Figure 6 depicts the effects on the HT29 (Figure 6A) and SW480 (Figure 6B) cell viability of AuNPsCS. Compared to AuNPsME (Figure 4) and AuNPsCE (Figure 5), AuNPsCS displayed a stronger inhibitory effect (*p <* 0.05) for 20–50 µg/mL, the same effect as AuNPsCE at 100 µg/mL, and the same effect as the other nanoparticles for 150 and 200 µg/mL in the HT29 cell line (Appendix A). For the SW480 cells, a lower effect than AuNPsME or AuNPsCE was observed for 20 µg/mL, a similar effect to AuNPsME at 50–100 µg/mL, and lower than AuNPsME at 150–200 µg/mL (Appendix A). Based on the calculated IC_50_ for HT29 (Figure 6C) and SW480 (Figure 6D), AuNPsCS are less effective against HT29 cells but have a better effect than AuNPsME in inhibiting SW480 cell proliferation. The representative pictures showed the strongest effects in the 100–200 µg/mL concentrations, agreeing with the results for cell viability.

AuNPsCS have been tested in colorectal cancer cell lines, exhibiting viabilities <80% in HT29 and SW480 cells at 1–200 µM concentrations, although HT29 cells displayed lower cell viabilities than SW480 [59], similar to the results presented in this research. It has been suggested that both HT29 and SW480 present a relatively high AuNPsCS uptake [59], suggesting that AuNPs-based nanoparticles could be an effective strategy aiming to deliver bioactive compounds in these cells.

In general, we can observe that the gold particles with the best results were those made with the ethanolic extract of castor oil with a dose-dependent effect, and it is well known that the size and capping agent of AuNPs plays an essential role in cellular viability and proliferation. We can observe this in the results because, compared to the chemical synthesis particles, the green ones presented a smaller particle size, and in the case of AuNPsCE this extract presents a greater antioxidant capacity and a higher content of phenolic compounds capping the surface of particles, which would lead to more significant activity. It is important to emphasize that the particles had a greater effect on SW480 cells that represent an early stage of colon cancer.

### 3.4. A Stronger ROS Generation Was Shown in the Ethanolic-Based AuNPs Compared to AuNPsCS

Reactive oxygen species (ROS) are a specific type of oxygen-containing reactive molecules that play an important role in different cellular processes and are essential for cell proliferation at basal levels. However, in high concentrations, they present a cytotoxic effect, causing necrosis or apoptosis and cell damage by ionizing DNA and other cellular molecules. This effect is important for different therapeutic applications, like the AuNPs anticancer effect that increases the production of ROS [60,61].

The effect of the evaluated nanoparticles on ROS generation in HT29 and SW480 cells is presented in Figure 7. Both green-synthesized nanoparticles showed a similar trend, where the highest concentration (200 µg/mL) surpassed the positive control (30 mM H_2_O_2_). Notably, AuNPsME (Figure 7A) displayed a higher ROS generation in 100 µg/mL than 150 µg/mL, a trend also shown for AuNPsCE (Figure 7C) but without differences with the 20–150 µg/mL range. Despite showing a similar behavior in SW480 cells, 200 µg/mL AuNPsME displayed a higher effect (Figure 7B) than AuNPsCE (Figure 7D). For AuNPsCS, similar effects were presented in HT29 (Figure 7E) to SW480 (Figure 7F).

The ability with which metallic nanoparticles can destroy cancer cells partially depends on the stimulation of ROS in the damaged components of the cells [62,63]. The generation of a ROS response in both cell lines after AuNPsME treatments supports other findings indicating that the sole ME treatment alters the production of superoxide anion in selected cell lines [53]. Verbascum calvum flower extracts were assayed against human lung A549 and breast MCF-7 cancer cells, showing antiproliferative effects attributed to ROS generation and pro-inflammatory effects [64]. Despite no reports for green synthesized AuNPsCE inducing ROS in colorectal cancer cells, phytochemicals from R. communis leaves such as p-coumaric acid have been linked to ROS generation, inducing further apoptosis and a fall in the mitochondrial membrane potential [57]. This effect has been observed in SW480 cells treated with phenolic-rich berries extracts [65]. *R. communis* leaves extracts induce ROS in LPS-stimulated murine RAW264.7 macrophages, acting as critical mediators of several pro-inflammatory responses, such as the canonical pathway or even NLRP3 inflammasome activation [34,39].

Overall, AuNPsCS were less effective in inducing ROS in the cancer cells, particularly at 200 µg/mL, compared to AuNPsME and AuNPsCE (Appendix A), which agrees with the reported low ability of AuNPs to produce ROS in immune cells [66], but effects depend on the cell line and the exposition time. Hence, AuNPs are considered ideal radiosensitizers in cancer radiotherapy since the high atomic number of Au enhances the radiation dose absorbed by tumors emitting secondary ionizing radiations, resulting in ROS production and increased oxidative stress [67].

### 3.5. Caspase-3/7 Activity Is Increased Due to AuNPs Treatments

Caspase-3 is a relatively small protein consisting of two subunits, 12- and 17-kDa subunits that contain three and five thiol functions, respectively, members of a family of Cysteine-Aspartic proteases, best known for their ability to mediate the cleavage of specific target proteins. Caspases are all produced initially as inactive zymogens (called procaspases) that are then subject to activation by a wide range of specific internal and/or external signals, such as a high production of ROS [67]. As observed in Figure 8, the treatments (IC_50_AuNPs) were less effective in inducing caspase 3/7 activity on HT29 (Figure 8A) than on SW480 cells (Figure 8B). For HT29, there were no differences between the positive control (30 mM H2O2) and AuNPs treatments, but the induction by AuNPsCE and AuNPsCS was higher (*p* < 0.05) than the untreated cells. Only AuNPsME and AuNPsCS displayed higher (*p* < 0.05) pro-caspase 3/7 effects than the positive control in SW480 cells, but all treatments were higher than the untreated cells. Obtained results indicated that, except for AuNPsME on HT29 cells, all treatments were able to induce Caspase 3/activity in both cell lines.

Within the radiosensitization provided by AuNPs, DNA damage induction and the development of pro-apoptotic mechanisms represent one of the most studied cytotoxic effects of the AuNPs-based nanoformulations in colon cancer [67]. Caspase-3 is a crucial driver of the intrinsic apoptotic pathway, potentially activated by ROS generation, leading to further cell death. Since the assayed AuNPs partially induced ROS and Caspase 3/7 activity in both cell lines, their relationship could be hypothesized. However, further in vitro and in vivo assays evaluating additional pro-apoptotic proteins are required to fully elucidate their mechanism of action, and not all the treatments were successful in inducing ROS, suggesting additional mechanisms inducing Caspase 3/7. It was hypothesized that gold nanoparticles synthesized with *V. thapsus* leaf extracts could induce apoptosis in several lung cancer cell lines [56], but no pro-apoptotic proteins were evaluated. *R. communis* leaves extracts-based nanoparticles evaluated in HepG2 and HeLa cell lines were presumably linked to DNA fragmentation due to ROS generation [39,42].

## 4. Conclusions

The antioxidant activity of the extracts can be related to the synthesis of gold nanoparticles since compounds capable of donating electrons can act as reducing agents of gold ions, capping the surface to stabilize them. Results suggested that gold nanoparticles obtained by green synthesis using *V. thapsus* flowers and *R. communis* leaves ethanolic extracts are suitable nanomaterials exhibiting antiproliferative effects through ROS generation and activation of caspase 3/7 activity in human HT29 and SW480 colorectal cancer cells. Compared to AuNPs obtained by chemical synthesis, AuNPsME and AuNPsCE displayed a more robust antiproliferative response in SW480 cells, and higher ROS generation in both cancer cell lines, with no differences in caspase 3/7 activation. The cytotoxic activity of AuNPs has a dose–response effect that depends on the particle size and capping compounds on the surface of AuNPs. The best cytotoxic activity was obtained with AuNPsCE, which presents the major phenolic compounds in the extract. We conclude that the capping compounds, morphology, and size are important factors in the cytotoxic activity of green gold nanoparticles, and this cytotoxicity is modulated by reactive oxygen species production and Caspases 3/7 activation. Despite the large size nanoparticles obtained with the chemical synthesis, they presented cytotoxic activity through both pathways studied, which can be related to the release of ions or particles that did not aggregate. To understand specific cancer-related mechanisms of action, studies on in vivo models are suggested.

## Figures and Tables

**Figure 1 pharmaceutics-14-02069-f001:**
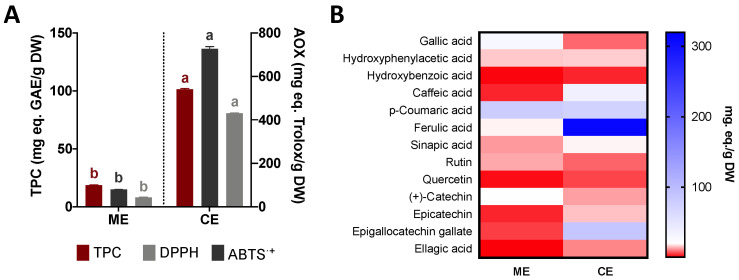
Polyphenolic composition of common mullein (*V. thapsus*) (ME) and castor bean (R. *communis*) (CE) extracts. (**A**) Content of total phenolic compounds (TPC) and antioxidant capacity (AOX), measured by ABTS and DPPH methods of *V. thapsus* (ME) and *R. communis* (CE) extracts. (**B**) Individual phenolic compounds identified by HPLC-DAD for ME and CE extracts. The results were expressed as the mean ± SD of three independent experiments in triplicates. Different letters in (**A**) (a, b) represent significant differences by Student’s *t*-test (*p <* 0.05) between ME and CE.

**Figure 2 pharmaceutics-14-02069-f002:**
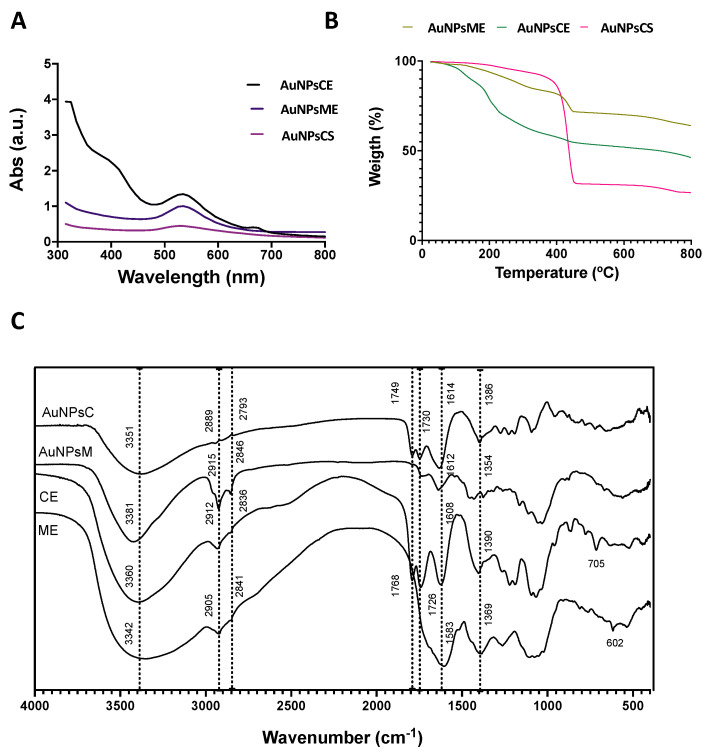
Synthesis and characterization of AuNPs. (**A**) Color change monitoring by UV-Vis spectra; (**B**) Thermogravimetric analysis (TGA) of AuNPs; (**C**) FT-IR spectra of the synthesized AuNPs. AuNPsCE: Gold nanoparticles synthesized with castor bean (*R. communis*) leaves extract; AuNPsCS: Gold nanoparticles produced by chemical synthesis; AuNPsME: Gold nanoparticles synthesized with common mullein (*V. thapsus*) flowers ethanolic extract; CE: Castor bean (*R.* communis) leaves ethanolic extract; ME: Common mullein (*V. thapsus*) flowers ethanolic extract.

**Figure 3 pharmaceutics-14-02069-f003:**
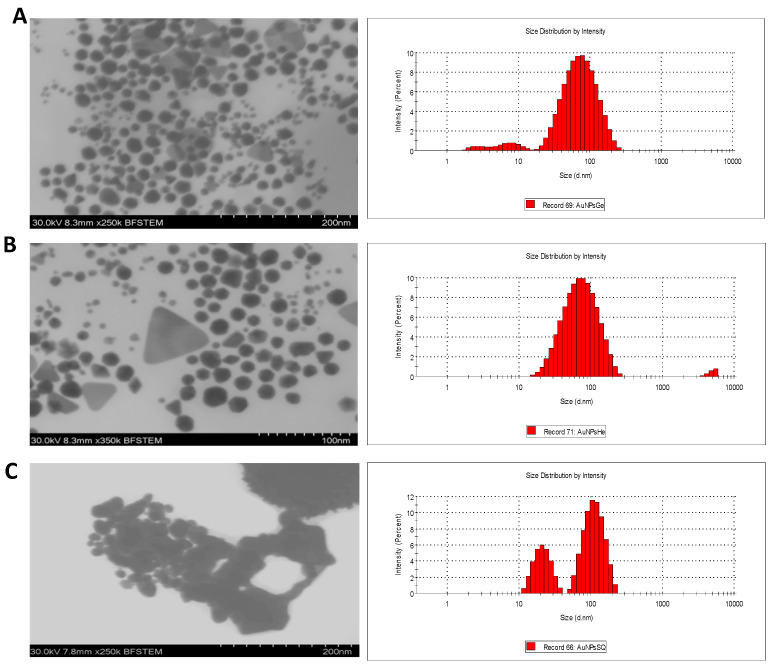
Scanning transmission electron microscope (STEM) and dynamic light scattering (DLS) analyses from (**A**) AuNPsME; (**B**) AuNPsCE; and (**C**) AuNPsCS. AuNPsCE: Gold nanoparticles synthesized with a castor (*R. communis*) leaves extract; AuNPsCS: Gold nanoparticles produced by chemical synthesis; AuNPsME: Gold nanoparticles synthesized with common mullein (*V. thapsus*) flowers ethanolic extract.

**Figure 4 pharmaceutics-14-02069-f004:**
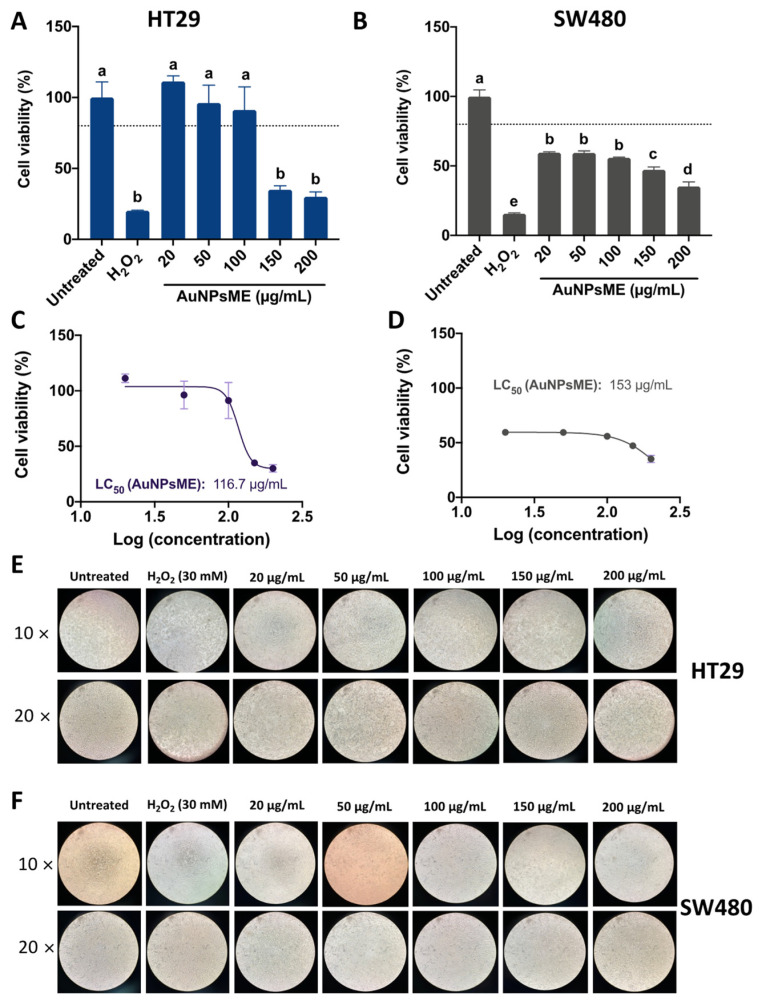
Assessment of HT29 and SW480 cell viability after treatment with common mullein (*V. thapsus*) flowers ethanolic extracts-gold nanoparticles (AuNPsME) treatments. (**A**) Evaluation of the impact of AuNPsME (20–200 µg/mL) and controls (untreated cells, 30 mM H_2_O_2_) in HT29 cells viability; (**B**) Effect of AuNPsME in SW480 cells viability; (**C**) Calculated IC_50_ of AuNPsME treatments on HT29 cells; (**D**) Calculated IC_50_ of AuNPsME treatments on SW480 cells; (**E**) Representative pictures (10× and 20×) of AuNPsME effects on HT29 cells; (**F**) Representative pictures (10× and 20×) of AuNPsME effects on SW480 cells. Hydrogen peroxide (H_2_O_2_)-treated cells (30 mM) were used as a positive control. Untreated cells (DMEM + 0.5% ASB) were used as a negative control. Pictures from untreated controls were the same as AuNPsCE and AuNPsCS. In Figure 4A,B, different letters (a, b, c, d, e) express significant differences (*p <* 0.05) by Tukey-Kramer’s test. The dashed line in (**A**,**B**) represents 80% cell viability.

**Figure 5 pharmaceutics-14-02069-f005:**
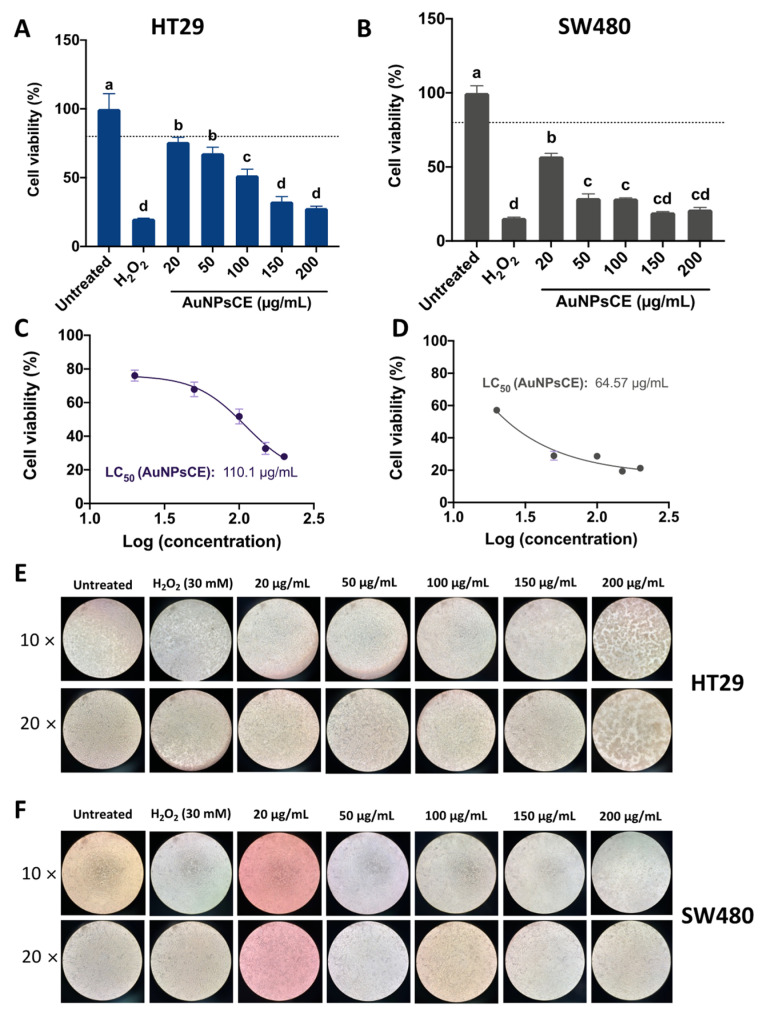
Impact of castor bean (*R. communis*) leaves ethanolic extracts-based gold nanoparticles (AuNPsCE) on HT29 and SW480 cells viability. (**A**) Effect of AuNPsCE in HT29 cells viability; (**B**) Influence of AuNPsCE in SW480 cells viability; (**C**) Quantified IC_50_ of AuNPsCE treatments on HT29 cells; (**D**) Quantified IC_50_ of AuNPsCE treatments on SW480 cells; (**E**) Representative pictures (10× and 20×) of AuNPsCE impact on HT29 cells; (**F**) Representative pictures (10× and 20×) of AuNPsCE impact on SW480 cells. Hydrogen peroxide (H_2_O_2_)-treated cells (30 mM) were used as a positive control. Untreated cells (DMEM + 0.5% ASB) were used as a negative control. Pictures from untreated controls were the same as AuNPsME and AuNPsCS. For Figure 5A,B different letters (a, b, c, d) express significant differences (*p <* 0.05) by Tukey-Kramer’s test. The dashed line in (**A**,**B**) represent 80% cell viability.

**Figure 6 pharmaceutics-14-02069-f006:**
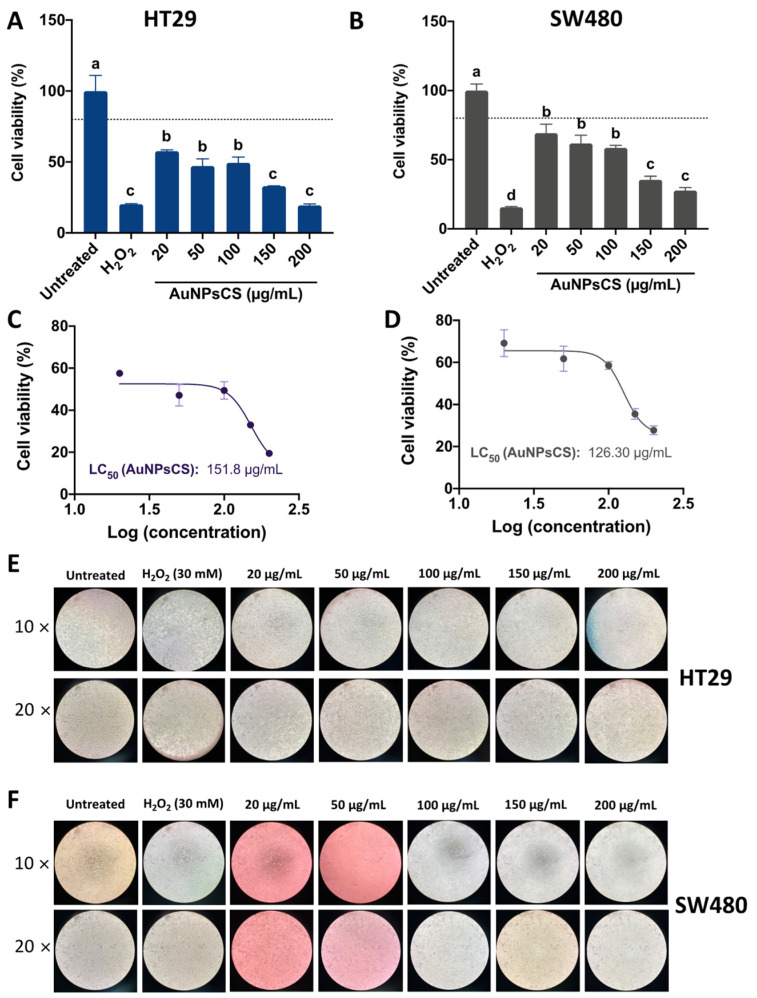
Impact of gold nanoparticles made by chemical synthesis (AuNPsCS) on HT29 and SW480 cells viability. (**A**) Effect of AuNPsCS in HT29 cells viability; (**B**) Influence of AuNPsCS in SW480 cells viability; (**C**) Quantified IC_50_ of AuNPsCS treatments on HT29 cells; (**D**) Quantified IC_50_ of AuNPsCS treatments on SW480 cells; (**E**) Representative pictures (10× and 20×) of AuNPsCS impact on HT29 cells; (**F**) Representative pictures (10× and 20×) of AuNPsCS impact on SW480 cells. Hydrogen peroxide (H_2_O_2_)-treated cells (30 mM) were used as a positive control. Untreated cells (DMEM + 0.5% ASB) were used as a negative control. Pictures from untreated controls were the same as AuNPsME and AuNPsCE. For Figure 6A,B, different letters (a, b, c, d) express significant differences (*p <* 0.05) by Tukey-Kramer’s test. The dashed line in (**A**,**B**) represent 80% cell viability.

**Figure 7 pharmaceutics-14-02069-f007:**
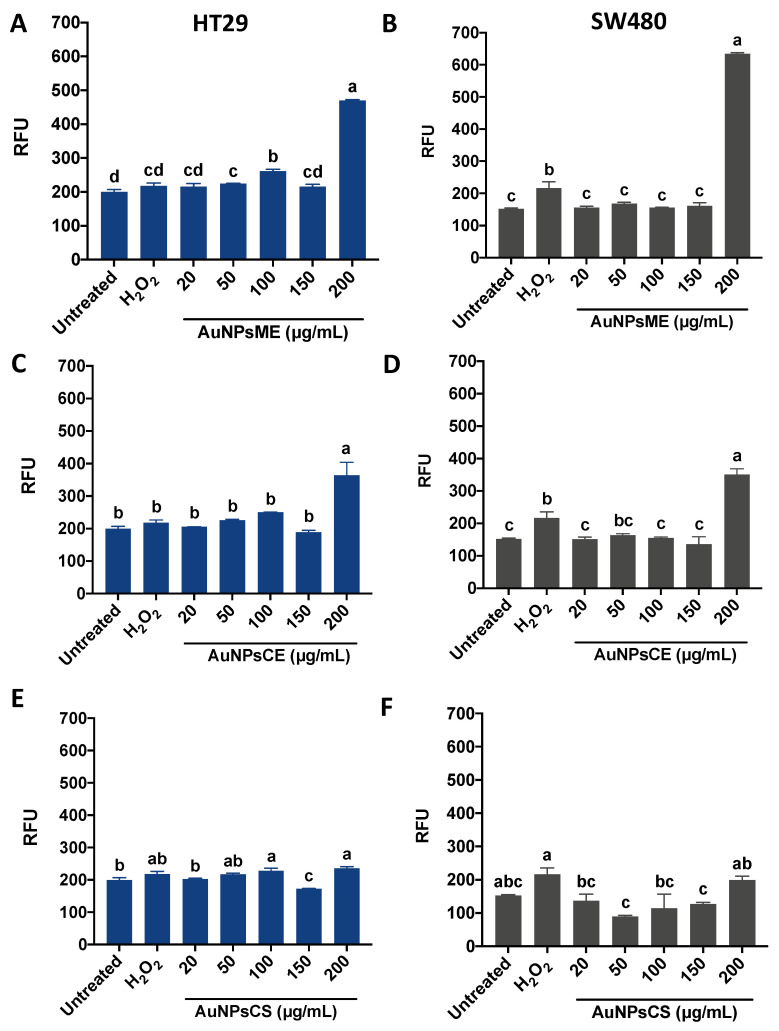
Intracellular reactive oxygen species (ROS) generation in HT29 and SW480 cells after treatments with common mullein (*V. thapsus*) ethanolic extracts-based gold nanoparticles (AuNPsME) (**A**,**B**); castor bean (*R. communis*) ethanolic extracts-based gold nanoparticles (AuNPsCE) (**C**,**D**); and gold nanoparticles made by chemical synthesis (AuNPsCS) (**E**,**F**). Different letters (a, b, c, d) in Figure 7A–F express significant differences (*p <* 0.05) by Tukey-Kramer’s test. AuNPsCE: Gold nanoparticles synthesized with a castor bean (*R. communis*) leaves extract; AuNPsCS: Gold nanoparticles produced by chemical synthesis; AuNPsME: Gold nanoparticles synthesized with common mullein (*V. thapsus*) flowers ethanolic extract; RFU: Relative fluorescence units. Hydrogen peroxide (H_2_O_2_)-treated cells (30 mM) were used as a positive control. Untreated cells (DMEM + 0.5% ASB) were used as a negative control.

**Figure 8 pharmaceutics-14-02069-f008:**
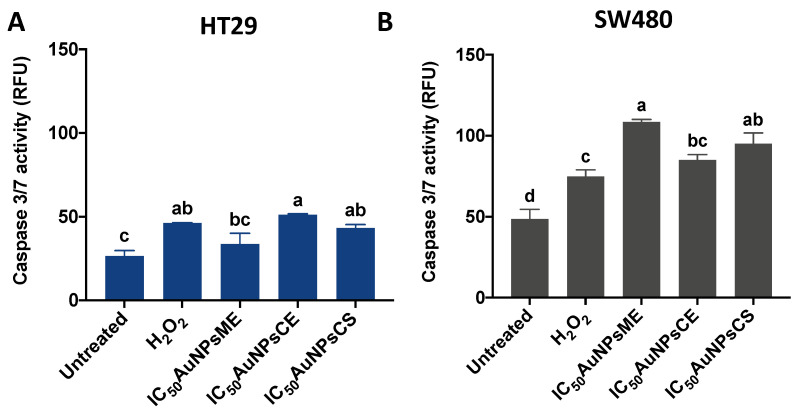
Evaluation of caspase 3/7 activity after treatment of HT29 (**A**,**B**) SW480 cells with the half inhibitory concentrations (IC_50_) of AuNPsME, AuNPsCE, and AuNPsCS. Different letters (a, b, c, d) in Figure 8A,B indicate significant differences (*p <* 0.05) by Tukey-Kramer’s test. AuNPsCE: Gold nanoparticles synthesized with a castor bean (R. communis) leaves extract; AuNPsCS: Gold nanoparticles produced by chemical synthesis; AuNPsME: Gold nanoparticles synthesized with common mullein (V. thapsus) flowers ethanolic extract; RFU: Relative fluorescence units. Hydrogen peroxide (H_2_O_2_)-treated cells (30 mM) were used as a positive control. Untreated cells (DMEM + 0.5% ASB) were used as a negative control.

## Data Availability

Not applicable.

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
