# Peer review of "Gold Nanoparticles Synthesized with Common Mullein (Verbascum thapsus) and Castor Bean (Ricinus communis) Ethanolic Extracts Displayed Antiproliferative Effects and Induced Caspase 3 Activity in Human HT29 and SW480 Cancer Cells"

_pharmaceutics, 2022, doi:10.3390/pharmaceutics14102069_

Round 1

Reviewer 1 Report

It was a manuscript about the synthesis and evaluation of Au nanoparticles through the green synthesis method using the extract of V. thapsus flowers (AuNPsME) and R. communis leaves (AuNPsCE). Here are some comments on this study that should be considered before publication:

1-      There are some grammatical mistakes in the text that should be corrected.

2-      “80-83 % A for 7 min, 83-60 % A for 5 min, 60-50 % A for 1 min, and 50-85 % A for 2 min” all of them are solvent A, so when will you use solvent B?

3-      Please introduce the manufacturer country of all the used instruments.

4-      “the antiox- 253 idant activity of the extracts can be related to the synthesis of gold nanoparticles since 254 having compounds capable of donating electrons can act as reducing agents of gold ions 255 and capping the surface to stabilize them.” It seems this part is not related to section 3.1.

5-      What do you understand from the results of the TGA test? You should mention that.

6-      You didn’t mention DLS in the method part.

7-      Why the hydrodynamic size of chemically synthesized nanoparticles was lower than STEM?

8-      “The PDI values that are more than 0.1 may imply polydisperse particle size distributions, which may be due to the variable compound composition of the extracts that cap the surface of AuNPs.” Please add a reference related to this sentence. Normally the monodispersity range is between 0.05 and 0.0.75.

9-      Please compare the MTT results of extract in the presence and absence of Au NPs.

10-   Section 3.5 needs to better description.

11-   The conclusion part needs to be improved.

Author Response

We greatly appreciate Reviewers #1 for their valuable comments and suggestions, so you can find the answers point by point in the attached file

Reviewer 2 Report

The manuscript describes the synthesis of gold nanoparticles using ethanolic extracts from two different medicinal plant flower/leaves: common Mullein and Castor Bean, respectively. The plant extracts and gold nanoparticles obtained are characterized by different instrumental techniques such as UV/vis, TGA, FTIR and STEM, among others. Then, the authors evaluate and discuss the antiproliferative effect and induced caspase activity of these green nanoparticles in two different human cancer cell lines, comparing the results obtained with those coming from chemically synthesized gold nanoparticles.

The paper seems to be interesting. However, several modifications/changes must be done prior to being published in Pharmaceutics journal, such as the ones listed below. That is why I recommend major revision of the manuscript.

General remarks

Some minor English grammar and spelling corrections should be done throughout the whole text. Besides, due to the high amount of acronyms employed in the manuscript, they should be carefully revised throughout the whole text as well.

Specific remarks

Keywords

I would try avoiding the use of acronyms in this section. Please, accompany it with the whole meaning.

Introduction section: Line 55-56: HAuCl4 is considered a salt, but the corresponding salt would be, for instance, KAuCl4. Please, revise this. The novelty and the focus of the paper should be better stated and clearly emphasized, since there are many different AuNPs obtained from plant extracts with similar antiproliferative activity as seen in literature. Hence, the novelty must be well introduced here in this section and in the abstract too. References are updated and I think they represent the state-of-the-art of the topic. Well done!

Materials and methods section: there are only a few reagents mentioned in this part. I think a full list should be included.

In section 2.3, the authors state they add quickly NaBH4 to the precursor solution to carry out the chemically driven synthesis. Perphaps if they perform the addition on the opposite way and slowly, they will obtain smaller AuNPs. Why did not they do that?

According to the assay perform, replicates were different in number: for example, ROS quantification by triplicate, Caspase activity by duplicate and MTT assay using six replicates. What is the purpose of using different number of replicates?

In section 2.5.3, line194, authors state a ‘Recently…’ A reference should be given here.

Results section: In Figure 1B, coloured legend is not defined.

Figure 2: colours of the different FTIR spectra are confusing. Please, change them.

Section 3.2.2: the number of significant ciphers should be used with special care. Besides, the number of decimal ciphers for the average size and standard deviation is excessive. I think they should be given only with one decimal cipher, since typically the resolution of the STEM instrumentation is 0,1 nm. Please, correct all these values according to the resolution of the STEM microscope.

Figure 3. Magnification of the STEM micrographs provided is too much high. In my opinion, higher magnification images should be included, around 100 nm or even less, in order to appreciate well the morphology and real size of the AuNPs. For example, Figure 3c seems to show a flower-like morphology for the AuNPs as authors state; however, perhaps with higher magnification, morphology can be appreciated in a better way, since in the actual micrograph, the flower-like structures seem to be agglomerations of smaller AuNPs. Moreover, instead of DLS histogram, a histogram from STEM would have been better, since they provide a much realistic value of the size distribution and average sizes. Authors talk about predominantly spherical morphology; however, when using plant extracts, polymorphism is typically obtained. Some explanation about that should be given in the text.

Conclusions

The actual section describes results but not conclusions. This section should be enhanced and include the main achievements according to novelty stated before. Something is missing in the first sentence.

References

Revise references: 4, 5, 45, 50, 53, 55 and 57. Some information is missing on them or repeated. Please, check. Furthermore, references 7 and 9 are the same cite. One of them must be removed and the rest of cites re-numbered properly.

Author Response

我们非常感谢Reviewers #2 的宝贵意见和建议,您可以在附件中逐点找到答案

Reviewer 3 Report

Gold Nanoparticles Synthesized with Common Mullein (Verbascum thapsus) and Castor Bean (Ricinus communis) Ethanolic Extracts Displayed Antiproliferative Effects and induced Caspase 3 Activity in Human HT29 and SW480 Cancer Cells

This paper describes the synthesis of gold nanoparticles using ethanolic extracts from the flowers of Verbascum thapsus and the leaves of Ricinus communis follow by the evaluation of the antitumoral activity against colon cancer cell lines. Although the topic of research is interesting there are several problems with this paper that should be address before publication. Some of them are indicated below:

General comments.

Authors should do an extensive revision of the manuscript, some important typos were found such as the use of NBH4 instead of NaBH4 (Page 6 line 289; Page 8 line 351) and should correct the use of the subindex (p.e Page 2 line 56, 91).

Authors should pay attention to the writing of species, using the full binomial name in italic in the first mention (Verbascum thapsus, Ricinus communis) and the abbreviation form thereafter (V. thapsus, R. communis).

Introduction.

Introduction is too brief and fails to convey the interest of research topic.

Also, the citations could be improved. For instance, in the first paragraph for the interest of gold nanoparticles they use only a specific articles on the green synthesis of gold nanoparticles with a plant instead of review articles from this topic.

Also, there are some parts on the introduction missing references. For instance, the paragraph regarding the methods of synthesis of gold nanoparticles and the interest of the green synthesis only has one reference.

Have been these species previously employ for the synthesis of nanoparticles?

Material and methods

Authors only tried one experimental condition. Did they not optimize the extraction conditions or the synthesis conditions? Why did they use different volumes from each extract instead of the same for a better comparison?

I think authors needs to correct the experimental section 2.5.1. Right now it is written that they test the extracts (“The media (DMEM+10 % FBS) 164 was then replaced with several concentrations of the extracts (20, 50, 100, 150, and 200 165 μg/mL extract) dissolved in DMEM”) and not the nanoparticles. While in the results its observed that they did not test the extracts.

Results and discussions

Regarding the results obtained for the synthesis of nanoparticles, in the TEM analysis, authors should include the nanotriangles (or other shapes) in the measurement of the size distribution, since they are present in the samples tested for the antiproliferative activity. Or did the authors purified the samples before the test to eliminate these bigger particles? They can’t have such small deviation with the difference in size observed in the images.

Regarding the chemical synthesized nanoparticles, why did the authors choose this method for the synthesis (Ref 20)? In this method they report the synthesis of AuNPs with mean size of 3 nm, while your green synthesized samples have a mean size of ~20 and ~37 nm. So, for a good and relevant comparison you should have tried to synthesized nanoparticles of a similar size, since it is a crucial factor on the activity of the nanoparticles.  Also, your chemical synthesized nanoparticles are very different from the ones in the reference article and obtained ~100 nm nanoparticles, which are much bigger than yours and they are not comparable.

Authors should analyzed the Z-potential of the nanoparticles, since the charge of the nanoparticles could be related to the activity and mechanism of action of the nanoparticles.

Regarding the antiproliferative assays:

Authors should analyzed the activity of the ethanolic extracts of Verbascum thapsus and Ricinus communis at the concentration present in the AuNPs samples as control, in order to stablish if the activity is due to the extracts, the nanoparticles, a synergistic activity, etc… This comparison seems more interesting that the comparison with the chemical synthesized nanoparticles.

Author Response

We greatly appreciate Reviewers #3 for their valuable comments and suggestions, so you can find the answers point by point in the attached file.

Round 2

Reviewer 1 Report

Thank you very much from the authors for their complete reply.

Reviewer 2 Report

After revising the new version of the manuscript, I consider that its quality has been increased and it is suitable for publication in its present form. However, please, consider this minor changes prior to publishing.

1) Figure 1B in the manuscript has not been changed as in the report notes sent to referee.

2) Please, revise as well sentence from lines 658 to 662. Something is missing there or perhaps the word 'which' after the colon should be removed.

Reviewer 3 Report

Authors had kindly respond to all my suggestions and comments.